# Overwintering Larval Cold Tolerance of *Sirex noctilio* (Hymenoptera: Siricidae): Geographic Variation in Northeast China

**DOI:** 10.3390/insects12020116

**Published:** 2021-01-28

**Authors:** Chengcheng Li, Jiahe Pei, Jiale Li, Xiaobo Liu, Lili Ren, Youqing Luo

**Affiliations:** 1Beijing Key Laboratory for Forest Pest Control, School of Forestry, Beijing Forestry University, Beijing 100083, China; lichengcheng311@163.com (C.L.); peijiahe@gmail.com (J.P.); ljl670509@163.com (J.L.); superliuxb@163.com (X.L.); 2Sino-French Joint Laboratory for Invasive Forest Pests in Eurasia, Beijing Forestry University—INRAE, Beijing 100083, China

**Keywords:** cold tolerance, supercooling point, lower lethal temperature, historical climate, habitat suitability, range expansion

## Abstract

**Simple Summary:**

The cold tolerance strategy of *Sirex noctilio* overwintering larvae was freeze-avoidance and their supercooling points adjusted seasonally to avoid icing damage. The cold hardiness between different populations varied on a spatiotemporal scale; specifically, the harsher the environment, the stronger the cold tolerance of overwintering *Sirex noctilio* larvae. A tunnel played a certain protective role but did not affect the low temperature parameters limiting the distribution of *S. noctilio*. We provide experimentally determined low temperature limiting parameters of *Sirex noctilio* and prove it has the opportunity to spread and colonize in other regions.

**Abstract:**

*Sirex noctilio* (Hymenoptera: Siricidae) is an invasive pest that has spread and established in many regions worldwide. However, its cold tolerance strategy is still unclear. We measured the supercooling point (SCP) and the lower lethal temperature (LLT) of overwintering *S. noctilio* larvae during three overwintering periods in four geographically separated populations in China. In addition, using the statistical analysis of the local historical temperature data, we also conducted comprehensive studies of *S. noctilio* cold tolerance variations and strategies. We measured the SCP of all samples as *S. noctilio* could survive at its SCP during a short period of exposure (<48 h) and its cold tolerance strategy was freeze-avoidance. The average SCPs of the groups in different spatiotemporal scales were significantly related to average temperature variation with most individuals exhibiting stronger cold hardiness at low ambient temperatures. *S. noctilio* exhibited a strong cold tolerance and it has the ability to withstand lower temperatures in cold environments. The geographic population showed a positive tendency as the ambient temperature decreased, which would increase its chance of developing in cold regions.

## 1. Introduction

*Sirex noctilio* Fabricius is an invasive woodboring wasp that attacks *Pinus* spp. (Pinaceae). It has become one of the most important pests in softwood plantations of the Southern Hemisphere [1,2,3,4,5]. In 2013, *S. noctilio* was found in Daqing, Heilongjiang Province in northeastern China, for the first time [6]. As reported [7], individuals of this species were successively found in Heilongjiang Province, Jilin Province, Liaoning Province and Inner Mongolia, indicating a strong ability to spread throughout the region. For invasive pests, temperature directly constrains the geographic distribution [8,9] and the spread [10]. The geographic distribution of invasive species in cold regions is limited by cold tolerance [11]. During the cold winter season, larval woodwasps enter the diapause within deep tunnels of host trees [12]. We measured the supercooling point (SCP) and low lethal temperature (LLT) for its population in China and described its thermal survival limits [13]. However, the cold tolerance strategy and climate variability of *S. noctilio* are not clear.

The climate variability hypothesis states that organisms exposed to a broader range of temperature variations exhibit greater tolerance to extreme temperatures [14]. Insects exposed to temperatures below the humoral melting point (MP) are at risk of fatally freezing their body fluids [15]. According to ice formation in bodily fluids and survival, there are three basic cold tolerance strategies: chilling intolerance, freeze-avoidance and freeze-tolerance [16]. The cold tolerance strategy of insect populations can be determined by comparing the SCP (the moment at which latent heat is released during icing) with the lower lethal temperature (e.g., LLT_99_, 99% of individuals die at that temperature) [17,18,19].

Different geographic habitats may cause differences in the cold tolerance of insects such as *Anoplophora chinensis* (Forster) and *A. glabripennis* (Motschulsky); see for instance [20,21,22,23,24]. Feng et al. assessed the variation in the cold tolerance of *A. glabripennis* in geographic populations and between seasons [20,21]. The results showed that the SCP and freezing point differed significantly among populations where the SCP of larvae in the Wulateqianqi (lower environment temperature) population was the lowest and highest in the Beijing (higher environment temperature) population [20]. At the same place and season, *A. glabripennis* may be better at maintaining metabolic activity at cold than *A. chinensis*. The distribution areas of *A. glabripennis* have lower environment temperatures than that of *A. chinensis* [22].

At present, there have been a few studies on the potential distribution of *S. noctilio* but most of their parameters were based on theoretical hypotheses rather than laboratory data [2,4]. We assumed that the cold tolerance and the time-space differences of *S. noctilio* in different populations would affect their potential distributions. We investigated the cold tolerance of *S. noctilio* by measured SCP of four geographic populations and combined these with LLTs. We then compared the cold tolerance of different geographical and seasonal populations. Furthermore, to examine the mechanisms affecting the temperature tolerance and distribution and, as a theoretical guide for the control of this invading pest, we discussed the capacity of *S. noctilio* to spread under temperature limits and assessed the cold tolerance parameters of larval *S. noctilio* based on the average and extreme temperatures of four different regions in China. We proposed the hypothesis that a lower level of cold tolerance capacity may limit the northward spread of *S. noctilio*. 

## 2. Materials and Methods

### 2.1. Sample Collection

Sample woodwasps were collected from four regions during three different phases of the overwintering period (pre-winter, mid-winter and post-winter) from October 2017 to March 2018 (Figure 1). Pre-winter means from October 2017 to November 2017, mid-winter means from December 2017 to January 2018 and post-winter means from February 2018 to March 2018. The specific temperatures of the experimental sites are shown in Table 1. The sample collection sites were in Hegang City, Heilongjiang Province (HG), Jinbaotun, Horqin Left Wing Rear Banner, Tongliao City, Inner Mongolia Autonomous Region (JBT), Yushu City, Jilin Province (YS) and Duerbote Mongolian Autonomous County, Daqing City, Heilongjiang Province (DM).

Infested pines (weak *Pinus sylvestris* var. *mongolica* Litv. with resin drops) were cut down and sawn into woodblocks of 1 m in length. The logs were wrapped in nets and transported to Beijing Key Laboratory for Forest Pest Control, College for Forestry, Beijing Forestry University (the average temperature in winter is −1.43 °C). Overwintering larvae were obtained from the logs. First, we split the logs by a coffin machine (LS7T-520, Shanghai Baiduan Industry and Trade Co., Ltd., Shanghai, China) and then collected live larvae of relatively the same body size and instar. We used flat-headed tweezers to transfer the larvae to 2 mL tubes for temporary storage.

### 2.2. Supercooling Point

The SCP of the larvae was measured as soon as they were removed from the wood with nearly 30 larvae measured for each overwintering phase. To conduct the measurements, we fixed the body of an individual to the thermistor probe of a four-way insect SCP test system (TP100, Jiangsu Senyi Economic Development Co.; Ltd.; Jiangsu, China) with a sealing film. The SCPs of the larvae from different overwintering stages were recorded and measured using a sub-cooler [15]. The samples were placed in a high and low temperature test chamber (GDW-100, Beijing Yashilin Testing Equipment Co.; Ltd.; Beijing, China) at −35 °C and the temperature recorder recorded the change of larvae body temperature (the cooling rate was approximately 1 °C × min^−1^). The latent heat was released at the moment of icing, which represented the SCP. Each treatment was conducted and processed in the same way.

### 2.3. Lower Lethal Temperature

Differences in the SCPs were found between regions in December 2017 and February 2018. Based on preliminary assays, most *S. noctilio* larvae could survive at temperatures lower than their SCP. The LLTs of total populations have been calculated before [13]. The larvae from Yushu City, Jilin Province (YS) were selected for the low temperature exposure experiment during the mid-winter period, which was chosen because the lowest SCP measurements indicated the peak of cold tolerance at that time [13]. 

Taking into account the actual cold temperatures, we set the lower range with eight temperature gradients (5 °C, 0 °C, −5 °C, −10 °C, −15 °C, −20 °C, −25 °C and −30 °C). The duration of exposure of the sample to low temperatures was varied to approximate various exposure times to those temperatures in the wild. The larval low temperature exposure experiment was 2 h, 4 h, 24 h and 48 h. Each exposure time was implemented for each of the eight temperature gradients. Three to five samples were processed for each treatment. Each treatment was repeated three times. After completing the treatment, the samples were placed in a standard environment (25 °C, dark:light = 24:0 h, relative humidity = 75%) for 2 h and the body of each larvae was poked with a probe to determine whether it was alive. The damage to insects caused by the low temperature and the ability of insects to resist low temperatures were evaluated by calculating the lower lethal temperature causing 50% death (LLT_50_) and 99% death (LLT_99_) [13,16].

### 2.4. Historical Temperature Data and Microclimatological Measurements

Historical temperature data (1981–2010) for the four sites were obtained from the National Meteorological Information Center (http://data.cma.cn). The data included extremely low temperatures, monthly average temperatures and daily average temperatures. At the sampling sites, the temperature inside the overwintering tunnels and the ambient temperature outside were measured in the stem of three *Pinus sylvestris,* which had been injured by *S. noctilio*. Data (±1 °C) were recorded every ten minutes for the whole winter by electronic thermometers (L91-1, Hangzhou Luge Technology Co. Ltd.; Hangzhou, China). Test samples were three *P. sylvestris*, which were injured by *S. noctilio*, at the same sampling sites. We placed the temperature probes inside and outside the bored tunnel of the stem and recorded the data every ten minutes for the whole winter. Considering the practicality and general applicability of our research results, we chose the historical temperature data as the reference for analyzing the cold tolerance of *S. noctilio*.

### 2.5. Data Analysis

The SCP of the larvae were compared among different overwintering phases by analysis of variance (ANOVA) to evaluate the cold tolerance of the population as well as the changes throughout the overwintering periods. The SCP data were used as a reference for subsequent experiments. The SCP data were imported into SPSS 23.0 (IBM, Inc.; New York, NY, USA) and divided into three categories according to the date of collection. A least significant difference (LSD) (*p* = 0.05) multiplex analysis was used to compare the differences in the SCPs of samples among different periods and the Games–Howell test (*p* = 0.05) was applied for pairwise comparisons of different regions. The LSD and Games–Howell tests were the post hoc tests used with the ANOVA. We followed the methods of Li et al., Feng et al. and Li et al. to analyze the SCP and MP data [13,20,23]. A test of the homogeneity of variance was performed for one-way ANOVA to meet the normality criteria better and a *t*-test was performed between each pair of samples to obtain *p*-values. The relationship between the MP and the SCP was analyzed using Unpaired *t*-test data (after a Bonferroni-test and Bonferroni-corrected, *p* < 0.05). LLT_50_ and LLT_99_ were calculated [13,18,19] and the latter was used to evaluate the damage to insects caused by low temperatures and the ability of the insects to tolerate the low temperature. Data analysis consisted of the Probit method implemented in regression analysis in SPSS 23.0. October to March’s historical climate data were analyzed (Graph pad prism 7, GraphPad Software, Inc.; California, CA, USA). Spearman’s method was used to analyze the correlation between the characteristic values of the ambient temperature and the SCP because the characteristic values of the ambient temperature were continuous and had a large order of magnitude difference compared with the SCP.

## 3. Results

### 3.1. Supercooling and Melting Points of Overwintering Larvae

The average larval SCPs of the whole overwintering phases (October 2017 to March 2018) was −20.77 ± 0.44 °C (mean ± SE) for all populations [13]. The SCP temperatures ranged from −2.7 °C to −32.5 °C (Figure 2).

There were significant differences between the average SCP of the overwintering period (F = 26.616; df = 2319; *p* < 0.001). The average SCP of overwintering larvae collected at the pre-winter period (October to November 2017) was relatively higher than the other two periods with a value of −18.37 ± 0.71 °C (mean ± SE). Subsequently, the SCP rapidly decreased to the minimum values. The average values of all populations in the mid-winter period were −24.27 ± 0.62 °C (mean ± SE) indicating that the cold tolerance capacity of the overwintering larvae was significantly enhanced during this period. After this period, the SCP of the overwintering larvae increased to −18.42 ± 0.73 °C (mean ± SE) during the post-winter period, clearly indicating that their cold tolerance was weaker than in the mid-winter period. 

The MPs (mean ± SE) of the three overwintering periods were −10.62 ± 0.70 °C, −16.06 ± 0.57 °C and −11.42 ± 0.59 °C, respectively (Figure 2). The mean MP and SCP correlations of all experimental samples were significant (t = 12.4; df =379; *p* < 0.0001). The MPs corresponded with the SCPs (r = 0.83, *p* < 0.01) (Figure 2).

### 3.2. Diversity of SCPs from Different Geographic Populations

During the pre-wintering period, the SCP of the JBT and YS populations were significantly higher than that of the other populations (t = −8.02; df = 61.02; *p* < 0.001) (Figure 2 and Figure 1: HG-Hegang, JBT-Jinbaotun, YS-Yushu, DM-Duerbote Mongolian). The SCPs of the DM population during the mid-wintering period were lower than that of the other populations (t = −3.50; df = 51.51; *p* < 0.001). The SCP values (mean ± SE) were −22.48 ± 0.92 °C for the HG population (n = 71), −18.03 ± 0.83 °C for the JBT population (n = 74), −19.34 ± 5.87 °C for the YS population (n = 94) and −24.11 ± 1.00 °C for the DM population (n = 80). Moreover, the differences in the SCPs with respect to collection sites were also observed (F = 16.63; df = 3, 318; *p* < 0.001), which were mainly divided into two categories: the northern regions (DM and HG) and the southern regions (JBT and YS) (Figure 2). 

### 3.3. LLTs of Overwintering Larvae during Mid-Winter

Low temperature stress-induced mortality of the overwintering larvae increased with a decreasing temperature. When the larvae were exposed to a low temperature for less than 4 h, most samples survived above −15 °C; when the duration of stress exposure was 24 or 48 h, most of the larvae survived above −5 °C (Figure 3). In contrast, when larvae were exposed to a low temperature for 4 h, less than 30% survived below −20 °C.

The semi-lethal temperature (LLT_50_) and 99% lethal temperature (LLT_99_) of the overwintering larvae increased with the extension of exposure time (Table 2, [13]). At 2 h exposure, −22.09 °C caused 50% mortality and −29.77 °C caused 99% mortality. After 48 h of continuous exposure, the LLT_50_ and LLT_99_ of the overwintering larvae were −2.06 °C and −20.40 °C, respectively. 

### 3.4. Historical Climate Data of Sample Collection Sites

Based on historical data (1981–2010), the temperatures of the overwintering period at YS, HG and DM were similar, with those at JBT being warmer than those at other sites (F = 2.94; df = 3, 1456; *p* = 0.03) (Figure 4a). The average temperatures (mean ± SE) during the overwintering period were −8.12 ± 0.58 °C in HG, −4.44 ± 0.55 °C in JBT, −7.13 ± 0.62 °C in YS and −8.18 ± 0.63 °C in DM. As shown in Figure 4, the climate condition varied within each habitat: (1) the temperature at YS decreased slowly; (2) the DM temperature was extremely low for a long period of time; (3) the curve of HG was gradual, with almost no extreme values (Figure 4). According to the two categories divided by the SCP, we found that the northern and southern sites significantly differed in their daily average temperature (t = −2.85; df = 1, 1458; *p* < 0.01) (Figure 4b). The minimum mean temperature from October to March was −8.3 °C in the northern site (HG) and −4.5 °C in the southern site (JBT). In addition, the historical all-time minimum was −36.8 °C (DM, January 2001) (Figure 4c). By real-time monitoring of the relationship between the temperature inside and outside the sampling tunnel and the temperature of the weather station (S1), it was found that the tunnel temperature exhibited slight fluctuations and delayed changes but there was no difference in the overall temperature (Pearson test, *p* > 0.05).

### 3.5. Relationship Between Historical Climate Data and the Cold Tolerance of S. noctilio Populations

The SCPs of each population corresponded to the sampling site temperatures including the monthly average temperature (r = 0.71, *p* < 0.05), the monthly average low (r = 0.73, *p* < 0.01) or high (r = 0.67, *p* < 0.05) temperature and the monthly extreme low (r = 0.72, *p* < 0.01) or high (r = 0.69, *p* < 0.05) temperature during the overwintering period. The MPs were slightly lower than the corresponding average monthly temperature. The mean SCP of either group was higher than the historical average environmental temperature suggesting that the ambient mean temperatures were higher than the mean SCP of *S. noctilio.* However, this does not exclude some extreme values to be lower than the SCP.

## 4. Discussion

### 4.1. Cold Tolerance Strategy of S. noctilio Overwintering Larvae

*Sirex noctilio* is widely distributed including the northernmost regions of the world and even in the Russian Federation, Canada and Norway [25]. In our results, the larvae of *S. noctilio* could not survive for extended periods at temperatures below their SCPs and their LLT_50_ values were similar to their SCPs. The cold tolerance strategy could be determined by cooling a group of individuals to a temperature close to their SCP.

Based on the average SCPs measured during different overwintering periods (Figure 2), we proposed that the SCPs of *S. noctilio* adjusted seasonally. The abovementioned phenomenon indicates that the overwintering larvae of *S. noctilio* have a strong tolerance to short-term extreme low temperatures but do not tolerate long-term low temperature stress. Freezing-intolerant species seasonally depress the SCP of their bodily fluids to increase their cold tolerance to remain unfrozen in the supercooling stage [15,26]. In addition to defining freeze-intolerant, we would also highlight that truly freeze-avoiding means they can survive all the way down to the SCP. The freeze-intolerant insects are chill susceptible, meaning they die from cold well above the SCP. Therefore, the cold tolerance strategy of the larvae of icing-resistant insects can be inferred by comparing ambient temperatures during the overwintering period with their SCPs [16]. During the overwintering period, the lowest average daily temperature at the site of sample collection was nearly within the range of the SCPs of the overwintering larvae. This finding further indicates that in their natural state, overwintering larvae avoid the freezing of their bodily fluids [17], i.e., *S. noctilio* larvae are freeze-avoiding.

### 4.2. In Relation to Historical Climate Data, the Cold Hardiness of Populations Varies on a Spatiotemporal Scale

For freezing-intolerant species, the SCP represents the lower limit at which survival can occur [26]; thus, the SCP is a suitable measure for judging a species’ cold tolerance. Based on our results, the temperature of JBT was significantly higher than that in the other regions followed by YS (Figure 4). Although the average temperature of YS during the whole wintering period was similar to that of HG and DM, its temperature suddenly dropped in the middle of the wintering period, which was also consistent with the situation shown by the SCP. The SCPs of each population corresponded to the sampling site temperature during the overwintering period (JBT > YS > DM > HG); the northern populations (HG and DM) were considerably more cold-hardy than were the southern populations (JBT and YS) as indicated by the SCP measurements. The average SCP of all populations during the overwintering period was consistent with the average site temperature during the overwintering period (pre-winter and post-winter > mid-winter). During the winter, consistent with the environmental temperature changes, the SCPs of geographically distinct populations first decreased and then increased (Figure 2 and Figure 4).

Nonetheless, the temperature was not the only factor limiting the larval cold tolerance [14]. According to the climate variability hypothesis, geographical latitude, seasonality in temperature and also elevation above sea level are thought to correlate with the animals’ temperature tolerance breadth [27,28] (see for instance [29,30,31]). In our study, the average SCPs of different populations were ranked in the same patens with the latitude of our collection sites. Although the SCPs of *S. noctilio* were similar among sites and periods, the overwintering larvae of *S. noctilio* exhibited stronger cold hardiness in colder environments. In addition, the genotype and/or phenotype may influence differences in the SCP by the site [7].

### 4.3. Low Temperature Parameters Limiting the Distribution of S. noctilio

Previous analyses of suitable areas were based on the known distribution of a species [2,4]. The model of Carnegie et al. [2] fitting *S. noctilio* potential distribution was based on climate data in the native range, which was suitable for a large-scale biogenic area prediction. Ireland et al. [4] performed a potential distribution analysis using laboratory data. The authors included more local data points when validating the model than did the previous authors. Their model contained more known distribution points of *S. noctilio* especially the points of China [6]. The latter was relatively more suitable for describing the current stage and detailed analysis. Some parts of the latter model might be further improved by our experiments. First, our results supported the principle of Ireland et al. [4] who used laboratory data rather than distribution area data only. Our results might be more appropriate than those of Madden [12] in distribution prediction for describing the current temperature index of *S. noctilio* in China [13]. Second, the overwintering habit of *S. noctilio* should be considered. *S. noctilio* overwinters in the xylem of plants and undergoes diapause [12,32] and the tunnels created by borers have a certain insulating effect that reduces the influence of extreme temperatures (Appendix A, [6,33]. Third, exposure to temperatures below the MP strongly affects survival [34]. Insects that are not resistant to freezing might die during periods prior to freezing and mortality increases with exposure to temperatures above the SCP; therefore, the MP is also a factor that should be considered [35]. 

## 5. Conclusions

We demonstrated that the overwintering larvae of *S. noctilio* are freeze-intolerant and can only survive for a short period at temperatures below the SCP corresponding to the extremely cold temperatures of several habitats in northeast China. We aimed to estimate the potential geographical range and spreading risk of *S. noctilio* based on our laboratory data. Given the rapid climate change and global trade as well as the global vulnerability and spread of *S. noctilio*, a comprehensive future study of the climate appropriate for this species is needed to inform conservation and management initiatives.

## Figures and Tables

**Figure 1 insects-12-00116-f001:**
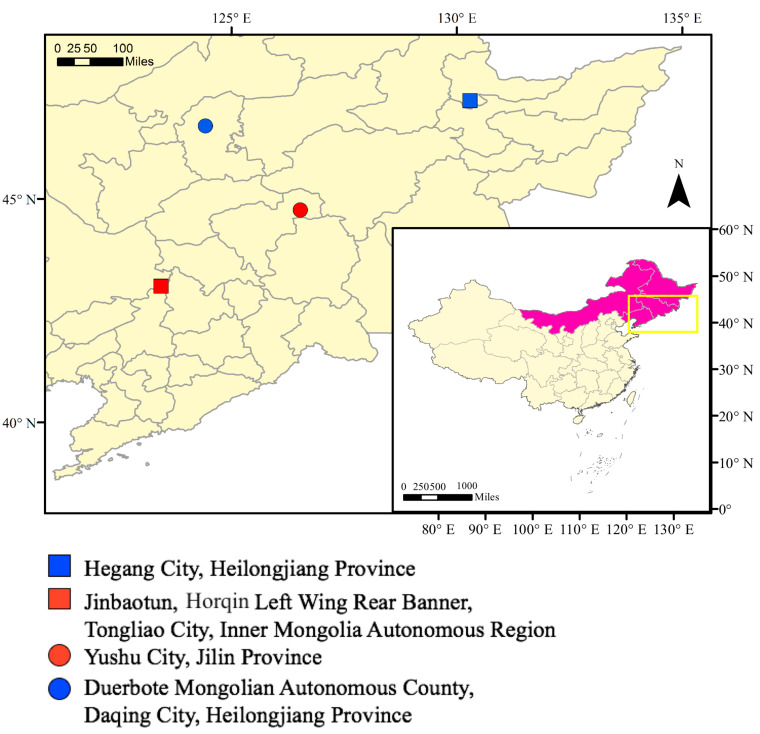
Sampling sites for *Sirex noctilio* in China showing the geographic range of the species by province. The inset map shows the location in the northeast of China.

**Figure 2 insects-12-00116-f002:**
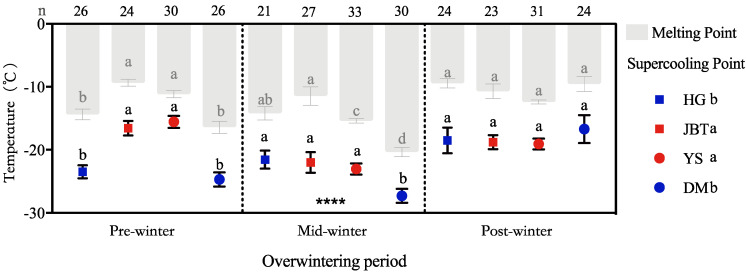
Supercooling points and melting points of *S. noctilio* larvae at different overwintering stages and sites. The bars are melting points (MPs) and the symbols are supercooling points (SCPs). Data points are means ± SEs and grey bars (one per column) are means of MPs with the 95% CI. Different letters in the same column indicate a significant difference with 95% confidence (Games–Howell test). **** means that the SCP of this group was tested to a difference of *p* < 0.0001 in Fisher’s LSD with others.

**Figure 3 insects-12-00116-f003:**
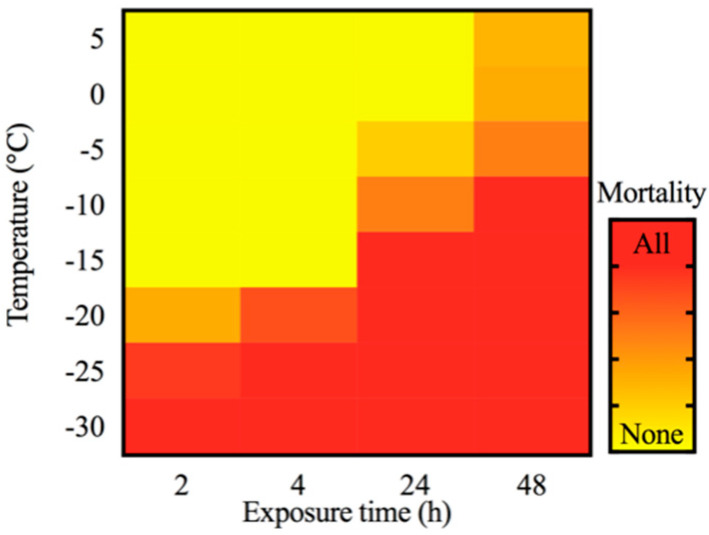
Mortality of overwintering larvae from the YS population under different exposure times during mid-winter. Red indicates a high mortality rate and yellow indicates there was low mortality or there was no mortality. The gradual transition from yellow to red corresponds to an increase in the mortality of samples from none (mortality 0%) to all (mortality 100%).

**Figure 4 insects-12-00116-f004:**
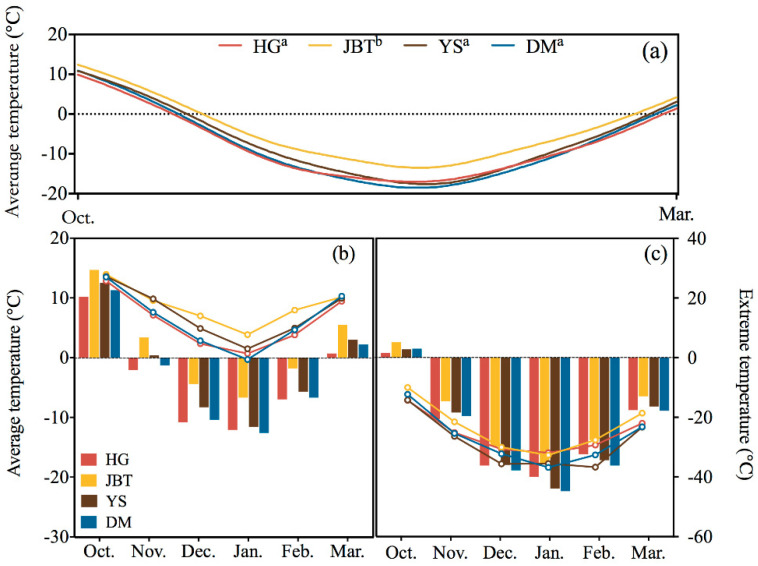
Historical climate data for the sampling sites (1981–2010). (**a**) Historical daily average climate data of the sampling sites (1981–2010). Only the overwintering periods were analyzed. Different letters in the same column indicate a significant difference with 95% confidence (LSD test). (**b**) Bars represent monthly average high temperatures, which correspond to the left Y axis. Lines show monthly extremely high temperatures, which correspond to the right Y axis. (**c**) Bars show monthly average low temperatures, which correspond to the left Y axis. Lines show monthly extremely low temperatures, which correspond to the right Y axis.

**Table 1 insects-12-00116-t001:** Specific temperatures of sampling sites. HG denotes Hegang City, Heilongjiang Province. JBT denotes Jinbaotun, Horqin Left Wing Rear Banner, Tongliao City, Inner Mongolia Autonomous Region. YS is Yushu City, Jilin Province. DM denotes Duerbote Mongolian Autonomous County, Daqing City, Heilongjiang Province.

Site	Annual Average Temp. (°C)	Extreme Low Temp. (°C)	Coldest Month Average Temp. (°C)	Coldest Month Average Low Temp. (°C)	Coldest Month Average High Temp. (°C)
HG	3.8	−31.9	−16.6	−20	−12.1
JBT	6.8	−32.7	−12.9	−17.8	−6.7
YS	5	−36.7	−17.1	−22	−11.6
DM	4.6	−36.8	−18	−22.4	−12.6

Historical temperature data (1981–2010) were gathered for collection sites from the National Meteorological Information Center (http://data.cma.cn/).

**Table 2 insects-12-00116-t002:** LLT_50_ (°C) and LLT_99_ (°C) of overwintering larvae with different exposure durations.

Exposure Time (h)	Sample Size	Slope + SE	LLT_50_ (°C)	50% Fiducial Limit		LLT_99_ (°C)	99% Fiducial Limit		Chi-Square
				Lower	Upper		Lower	Upper	
2	77	−0.30 + 0.09	−22.09	−24.52	−19.75	−29.77	−39.79	−26.56	0.67
4	79	−0.41 + 0.14	−19.23	−21.45	−16.94	−24.84	−35.79	−22.25	0.56
24	80	−0.21 + 0.05	−10.00	−12.49	−7.51	−21.07	−30.17	−17.21	0.83
48	81	−0.16 + 0.04	−3.53	−6.25	−0.41	−17.89	−28.57	−13.28	0.92

## Data Availability

The data presented in this study are available on request from the corresponding author. The data are not publicly available due to unfinished related ongoing further studies and manuscript.

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
