# Peer review of "Overwintering Larval Cold Tolerance of Sirex noctilio (Hymenoptera: Siricidae): Geographic Variation in Northeast China"

_insects, 2021, doi:10.3390/insects12020116_

Round 1
Reviewer 1 Report
This manuscript described an experiment which tested the potential difference of cold tolerance of Sirex noctilio from several populations. The results seem to suggest that there is a difference between the northern and southern populations representing colder and warmer climates, respectively and changes in freeze tolerance throughout the overwintering period. The experimental design and seems sound. The manuscript has parts of it that need better organization and the introduction and discussion mainly focus on this species and do not put the research in a broader context. I list my suggestions and concerns below in the order of appearance in the manuscript.
Abstract
1) Remove the subheadings (Background and Objectives, Materials and Methods …). This should be a paragraph that gives an overview of the study and its findings. Remove the actual numbers (lines 16-17) but just state the overall findings
Introduction
2) The introduction needs the be better organized. Consider using the front half of the introduction to set the stage for the experiment and the last portion to give an overview of what this study is testing. This structure will allow the authors to incorporate the existing literature more effectively (Such as other species that show geographical variation) and discuss the predictions of this study in a more organized fashion. As written, this part of the manuscript seems very disjointed.
3) Lines 44-45: Please add more examples in addition to A. chinensis.
4) Line 52: Please replace “have impacts on” with “affect”
Materials and Methods
5) Lines 61-62: Please state the time represented by the three periods. This information is provided later, but it should be stated in the methods.
6) Lines 68-70 repeat lines 63-65 and should be removed
7) Line 102: Why were only larvae from YS used? Why not all four populations? If your prediction in this study was there were differences in populations, representatives from all four populations should be tested here as well.
8) Line 127 please remove “: although”
9) Line 131: You should consider calling the section “data analysis” rather than “statistics”
10) Lines 132-146 (Statistics section): Given that SC and FP were tested it would seem you used a MANOVA not an ANOVA. It unclear was why multiple T-tests were used. The use of multiple T-tests would require a table-wide test such a Bonferroni test to ensure you do not get a false positive. Please make it cleared that the LSD and Games Howell tests were the post hoc tests used with the MANOVA. Why was a Spearman’s Test used? This is a nonparametric test. The other tests were parametric. Please clarify this decision in this section.
11) Section 3.1: This section is not very clearly organized and needs to be restructured so that the results are clear to the reader. For example. Line 150, “There were significant differences in the average SCP…” What exactly were the differences? Line 161, “was relatively high” compared to what? Lines 168-169 “… all experimental samples were significant”, it is unclear what this means. State which samples are significantly higher or lower than others.
12) Line 201: I think you meant Figure 4 here.
Discussion:
13) As mentioned above, more examples of other species would help enhance the narrative in the discussion throughout.
14) line 229-230: Remove the last sentence. This was already stated in the introduction
15) Line 252: I think you meant Figure 4 here
16) Lines 253-258: This paragraph needs to be reorganized so that it flows better.
17) Lines 285-290: This information does not belong in the conclusion and should be removed. It adds nothing the paper.
Figures
18) Figure 1: Was there a significance of using circles and squares? If so, please state that in the legend. I suggest you mention that the blue are northern populations and red southern populations in the legend.
19) Figure 3: in the key showing the gradient and Min and Max, please add percentages or ranges of percentages. As presented, this information seems very vague.
20) Figure 4: Avoid using red and green in the same figure. It is hard are red-green colorblind people. You could replace the green with purple.
Author Response
Response to Reviewer 1 Comments
Dear Reviewer:
Thanks a lot for your affirmation to our research. We tried our best to modify the "introduction" and "results" you mentioned. The specific changes are as follows:
Abstract
1) Remove the subheadings (Background and Objectives, Materials and Methods …). This should be a paragraph that gives an overview of the study and its findings. Remove the actual numbers (lines 16-17) but just state the overall findings
Thanks for your suggestion. We also reviced several minor points as shown in light blue in the manuscript. We removed the subheadings (Line 18-30). We removed the actual numbers, and rewrite the sentence as “We measured the SCP of all samples, as S. noctilio could survive at its SCP during a short period of exposure (<48 h), its cold tolerance strategy was freeze-avoidance.” (Line 23-25).
Introduction
2) The introduction needs the be better organized. Consider using the front half of the introduction to set the stage fod’sr the experiment and the last portion to give an overview of what this study is testing. This structure will allow the authors to incorporate the existing literature more effectively (Such as other species that show geographical variation) and discuss the predictions of this study in a more organized fashion. As written, this part of the manuscript seems very disjointed.
Thanks for your suggestion. We reorganized the structure of introduction: Sirex noctilio Fabricius is an invasive woodboring wasp that attacks Pinus spp. (Pinaceae). It has become one of the most important pests in softwood plantations of the Southern Hemisphere [1-5]. S. noctilio has a strict symbiotic relationship with its symbiotic-fungi Amylostereum areolatum (Fr.) Boidin [5]. In 2013, S. noctilio was found in Daqing, Heilongjiang Province in northeastern China, for the first time [6]. As reported [7], individuals of this species were successively found in Heilongjiang Province, Jilin Province, Liaoning Province, and Inner Mongolia, indicating a strong ability to spread throughout the region. For invasive pests, temperature directly constrains the geographic distribution [8, 9] and the spread [10]. The geographic distribution of invasive species in cold regions is limited by cold tolerance [11]. During the cold winter season, larval woodwasps enter the diapause within deep tunnels of host trees [12]. We had measured the supercooling point (SCP) and low lethal temperature (LLT) for its population in China, described its thermal survival limits [13]. However, the cold tolerance strategy and climate variability of S. noctilio are not clear.
The climate variability hypothesis states that organisms exposed to a broader range of temperature variations exhibit greater tolerance to extreme temperatures [14]. Insects exposed to temperatures below the humoral melting point (MP) are at risk of fatally freezing their body fluids [15]. According to ice formation in bodily fluids and survival, there are three basic cold tolerance strategies: chilling intolerance, freeze-avoidance, and freeze-tolerance [16]. The cold tolerance strategy of insect populations can be determined by comparing the SCP (the moment at which latent heat is released during icing) with the lower lethal temperature (e.g., LLT99, 99% of individuals die at that temperature) [17-19].
Different geographic habitats may cause differences in cold tolerance of insects [see, for instance (20-24)]. Such as Anoplophora chinensis (Forster) and A. glabripennis (Motschulsky), Feng et al. assessed cold tolerance variations between geographical population and seasons of A. glabripennis [20, 21]. The results showed that the SCP and freezing point differed signiÞcantly among populations, where the SCP of larvae in the Wulateqianqi (lower environment temperature) population was the lowest and highest in the Beijing(higher environment temperature) population [20]. At same place and season, A. glabripennis may be better at maintaining metabolic activity at cold than A. chinensis, the distributin areas of A. glabripennis have lower environment temperature than that of A. chinensis [22].
At present, there have been a few studies on the potential distribution of S. noctilio, but most of their parameters were based on theoretical hypotheses rather than laboratory data [2,4]. We assumed that the cold tolerance and the time-space differences of S. noctilio in different populations would affect their potential distributions. And our research is the first study focusing on the cold tolerance and its climate variability of S. noctilio. We investigated the cold tolerance of S. noctilio by measured the supercooling point (SCP) of 4 geographic populations and combined with LLTs. Then we compared the cold tolerance of different geographical and seasonal populations. Furthermore, to examine mechanisms affecting the temperature tolerance and distribution, and as a theoretical guide for the control of this invading pest, we 1) discussed the capacity of S. noctilio to spread under temperature limits; 2) assessed the cold tolerance parameters of larval S. noctilio based on the average and extreme temperatures of four different regions in China. We proposed the hypothesis that a lower level of cold tolerance capacity may limit the northward spread of S. noctilio.
3) Lines 44-45: Please add more examples in addition to A. chinensis.
Thanks for your suggestion. We added it at L55-59 as “Different geographic habitats may cause differences in cold tolerance of insects [see, for instance (20-24)]. Such as Anoplophora chinensis (Forster) and A. glabripennis (Motschulsky), Feng et al. assessed cold tolerance variations between geographical population and seasons of A. glabripennis [20, 21]. The results showed that the SCP and freezing point differed signiÞcantly among populations, where the SCP of larvae in the Wulateqianqi (lower environment temperature) population was the lowest and highest in the Beijing(higher environment temperature) population [20]. At same place and season, A. glabripennis may be better at maintaining metabolic activity at cold than A. chinensis, the distributin areas of A. glabripennis have lower environment temperature than that of A. chinensis [22].”.
4) Line 52: Please replace “have impacts on” with “affect”
Thanks for your suggestion. We replaced “have impacts on” with “affect” (Lines 62).
Materials and Methods
5) Lines 61-62: Please state the time represented by the three periods. This information is provided later, but it should be stated in the methods.
Thanks for your suggestion. We changed it, and stated it at Lines 79-80.
6) Lines 68-70 repeat lines 63-65 and should be removed
Thanks for your suggestion. We deleted Lines 68-70.
7) Line 102: Why were only larvae from YS used? Why not all four populations? If your prediction in this study was there were differences in populations, representatives from all four populations should be tested here as well.
Thanks for your suggestion. The larvae from Yushu City, Jilin Province (YS) were selected for the low-temperature exposure experiment during the mid-winter period, which was chosen because the lowest SCP measurements indicated the peak of cold tolerance at that time.
8) Line 127 please remove “: although”
Thanks for your suggestion. We removed it.
9) Line 131: You should consider calling the section “data analysis” rather than “statistics”
Thanks for your suggestion. We changed it to “data analysis” (Lines 140).
10) Lines 132-146 (Statistics section): Given that SC and FP were tested it would seem you used a MANOVA not an ANOVA. It unclear was why multiple T-tests were used. The use of multiple T-tests would require a table-wide test such a Bonferroni test to ensure you do not get a false positive. Please make it cleared that the LSD and Games Howell tests were the post hoc tests used with the MANOVA. Why was a Spearman’s Test used? This is a nonparametric test. The other tests were parametric. Please clarify this decision in this section.
Thanks for your suggestion. Our data analysis method was based on the following literatures:
- Li, C.; Wang, L.; Li, J.; Gao, C.; Luo, Y.; Ren, L. Thermal survival limits of larvae and adults of Sirex noctilio (Hymenoptera: Siricidae) in China. PLoS ONE. 2019, 14(6): e0218888.
- Yuqian, F.; Lili, X.; Bing, T.; Jing, T.; Jinlin, W.; Shixiang, Z. Cold hardiness of asian longhorned beetle (coleoptera: cerambycidae) larvae in different populations. Entomol. 2014, 5, 1419-26.
- Li, J.; Shi, J.; Xue, Y.; Mao, H.; Luo, Y. Major physiological adjustments in freezing-tolerant grey tiger longicorn beetle (Xylotrechus rusticus) during overwintering period. Fore. Res. h, 2014, 25(3), 653-659.
According to your suggestion, we have revised sentences in this section:
The SCP of the larvae were compared among different overwintering phases by analysis of variance (ANOVA) to evaluate the cold tolerance of the population as well as the changes throughout the overwintering periods. The SCP data were used as a reference for subsequent experiments. The SCP data were imported into SPSS 23.0 (IBM, Inc.) and divided into three categories according to the date of collection. A test of homogeneity of variance was performed for one-way ANOVA to better meet normality criteria, and a t test was performed between each pair of samples to obtain P-values. The relationship between FP and SCP is analysed using "Unpaired t-test data" (after Bonferroni-test and Bonferroni-corrected, p<0.05). A least significant difference (LSD) (P=0.05) multiplex analysis was used to compare differences in the SCPs of samples among different periods, and the Games-Howell test (P=0.05) was applied for pairwise comparisons of different regions. LSD and Games Howell tests were the post hoc tests used with the ANOVA. LLT50 and LLT99 were calculated [13,17-18] and the latter was used to evaluate the damage to insects caused by low temperature and the ability of the insects to tolerate the low temperature. Data analysis consisted of the Probit method implemented in regression analysis in SPSS 23.0. The historical climate data for October to March were analyzed (Graph pad prism 7, GraphPad Software, Inc). According to the ambient temperature characteristic values is continues, there is huge difference of orders of magnitude the SCPs, the correlations between them using Spearman’s method.
11) Section 3.1: This section is not very clearly organized and needs to be restructured so that the results are clear to the reader. For example, Line 150, “There were significant differences in the average SCP…” What exactly were the differences? Line 161, “was relatively high” compared to what? Lines 168-169 “… all experimental samples were significant”, it is unclear what this means. State which samples are significantly higher or lower than others.
Thanks for your suggestion. We rewrite this section:
The larval average SCPs of the whole overwintering phases (October 2017 to March 2018) was -20.77 ± 0.44°C (mean ± SE) for all populations [13]. The SCP temperatures ranging from -2.7°C to -32.5°C (Figure 2).
There were significant differences between the average SCP of overwintering period (F = 26.616; df = 2,319; P < 0.001). The average SCP of overwintering larvae collected at the pre-winter period (October to November 2017) was relatively higher than the other two periods, with a value of -18.37 ± 0.71°C (mean ± SE). Subsequently, the SCP rapidly decreased to the minimum values, the average values of all populations in mid-winter period were -24.27 ± 0.62°C (mean ± SE), indicating that the cold tolerance capacity of the overwintering larvae was significantly enhanced during this period. After this period, the SCP of the overwintering larvae increased to -18.42 ± 0.73°C (mean ± SE) during the post-winter period, clearly indicating that their cold tolerance was weaker than it in the mid-winter period.
The freezing points (mean ± SE) of the three overwintering periods were -10.62 ± 0.70°C, -16.06 ± 0.57°C and -11.42 ± 0.59°C, respectively (Figure 2). The mean FP and SCP correlations of all experimental samples were significant (t = 12.4; df =379; P < 0.0001). The freezing points corresponded with the SCPs (r = 0.83, P < 0.01) (Figure 2).
12) Line 201: I think you meant Figure 4 here.
Thanks for your suggestion. You were right, we changed it at line 211.
Discussion:
13) As mentioned above, more examples of other species would help enhance the narrative in the discussion throughout.
Thanks for your suggestion. We added more examples, such as other species, situations and even the heat-tolerance and symbiont-fungi.
Nonetheless, the temperature is not the only factor limiting larval cold tolerance [14]. According to the climate variability hypothesis and instance, geographical latitude, seasonality in temperature, and also elevation above sea level are thought to correlate with the animals’ temperature tolerance breadth [26-27, see for instance (28-30)]. After all, the average SCPs of different populations were ranked in the same order as the latitude of the collection site. What’s more, there were four experimental sites, different transport and storage conditions might have had an impact on the SCP [31]. Although the SCPs of S. noctilio were similar among sites and periods, the overwintering larvae of S. noctilio exhibited stronger cold hardiness in colder environments. In addition, genotype and/or phenotype may influence differences in SCP by the site
14) line 229-230: Remove the last sentence. This was already stated in the introduction
Thanks for your suggestion. We removed it.
15) Line 252: I think you meant Figure 4 here
Thanks for your suggestion. Here we meant to discussed the relation of the environmental temperature and the SCP of geographically distinct populations, we added Figure 4a at Line 265.
16) Lines 253-258: This paragraph needs to be reorganized so that it flows better.
Thanks for your suggestion. We recognized this paragraph as:
Nonetheless, the temperature is not the only factor limiting larval cold tolerance [14]. According to the climate variability hypothesis and instance, geographical latitude, seasonality in temperature, and also elevation above sea level are thought to correlate with the animals’ temperature tolerance breadth [26-27, see for instance (28-30)]. After all, the average SCPs of different populations were ranked in the same order as the latitude of the collection site. What’s more, there were four experimental sites, different transport and storage conditions might have had an impact on the SCP [31]. Although the SCPs of S. noctilio were similar among sites and periods, the overwintering larvae of S. noctilio exhibited stronger cold hardiness in colder environments. In addition, genotype and/or phenotype may influence differences in SCP by the site.
17) Lines 285-290: This information does not belong in the conclusion and should be removed. It adds nothing the paper.
Thanks for your suggestion. We removed it.
Figures
18) Figure 1: Was there a significance of using circles and squares? If so, please state that in the legend. I suggest you mention that the blue are northern populations and red southern populations in the legend.
Thanks for your suggestion. They were no significance, just for figure out different populations and correspond the same populations to Figure 2.
19) Figure 3: in the key showing the gradient and Min and Max, please add percentages or ranges of percentages. As presented, this information seems very vague.
Thanks for your suggestion. We added the information at Line 202-203 as “The gradual transition from yellow to red corresponds to an increase in the mortality of samples from none (mortality 0) to all (mortality 100%).”
20) Figure 4: Avoid using red and green in the same figure. It is hard are red-green colorblind people. You could replace the green with purple.
Thanks for your suggestion. We replaced the green with brown.Please see the attachment.

Reviewer 2 Report
Overwintering Larval Cold Tolerance of Sirex noctilio (Hymenoptera: Siricidae): Geographic Variation in China
Chengcheng Li, Jiahe Pei, Jiale Li, Xiaobo Liu, Lili Ren and Youqing Luo
This is a valuable and well done study on the thermal traits of Sirex noctilio determining its overwintering success in temperate and cold climates. Overall, the manuscript is well written (though clarifications are needed), methods applied are comprehensive (though not in all cases described in Methods), and statistical analysis seems appropriate. With great interest I noticed that the authors also took into account time of exposure to low temperatures as an important factor in their analysis.
Nevertheless, I have to recommend a "minor major" revision before it can go to print.
These are my main suggestions for improvement:
TERMINOLOGY:
I suggest the authors use the term "Freezing point" as a synonym of the insects' MELTING POINT throughout the paper and in Fig. 2. This is misleading and incorrect, and should be avoided, especially in the view of non-specialist readers (see [and quote] e.g. Zachariassen K E (1985) Physiology of cold tolerance in insects. Physiol Rev 65, 799-832).
The overwintering insects freeze at the supercooling point (SCP)! The ice crystals formed AFTER freezing (below the SCP) melt at the higher MELTING POINT (MP) during rewarming (thermal hysteresis).
I am missing a description in Methods how these MPs (gray bars in Fig. 2) were determined!
DISCUSSION
The discussion can stand some clarifications here and there.
OTHER:
Include Fig. S1 in the main paper! To me this is a quite interesting and informative recording on the relation of microclimatological thermal conditions between the overwintering hibernaculi and the outside (and historical) temperatures.
Rewrite its legend to (L295-296):
"Figure 1. Mid-winter temperature measured inside and outside the overwintering tunnel, and of the nearest local weather station (Historical)."
L28-29: "…that attacks Pinus spp. (Pinaceae). It has become one of the most …
L45: Maybe quoting here (or elsewhere) a recent paper from this journal might be helpful: Käfer et al. (2020) Temperature tolerance and thermal environment of European seed bugs. Insects 11, 197.
L66-67: The legend text is not correct. Cange it to:
"Figure 1. Sampling sites for S. noctilio in China, showing the geographic range of the species by province. The inset map shows the location in the northeast of China."
L68-70: DELETE!
L98-99: CHANGE (here and elswhere) "freezing point (FP)" to "melting point (MP)" (see above)
L105-112: CLARIFY: The experimental regime is unclear to me. Eight temperature "gradients"?
Did you cool the larvae to the eight temperatures (or just put them into chambers with those temperatures) and leave them there for the indicated periods of time?
L118: CHANGE HEADING:
"2.4. Historical temperature data and microclimatological measurements"
L121-125: CONDENSE and REWRITE THIS PAPAGRAPH!
"At the sampling sites, the temperature inside overwintering tunnels and the ambient themperature outside were measured in the stem of three Pinus sylvestris which had been injured by S. noctilio. Data (± 1°C) were recorded every ten minutes for the whole winter by electronic thermometers (L91-1, Hangzhou Luge Technology Co. Ltd.; China)."
L125-130: TRANSFER THIS PART TO RESULTS!
L136: "… date of collection."
L137: "… a t test was …"
L149‑150: avoid separation of minus sign from number at line break!
L152: insert space after °C
L167-170: CHANGE (here and elswhere) "freezing points (FP)" to "melting points (MP)" (see above)
L168-170: UNCLEAR!
correlations with what? The MPs correlated with the SCPs (r = 0.83, P < 0.01) (Figure 2).
L185: "In contrast, …"
L197; Fig. 3: It would be helpful to have the (color) classes shown in the figure also appearing in the color scale (Mortality: None to All)
L195, Table 2: prevent dividing (breaking) of words!
L199 (and elsewhere): WRITE " (mean ± SE) "
L201: you mean "… Figure 2, …" or "… Figure 4, …"?!
L201-203: THIS IS UNCLEAR, please CLARIFY (cannot see this in Fig. 2 or Fig. 4 clearly)!
L223: "…mean SCP of S. noctilio. However, this does not exclude some extreme values to be lower than the SCP."
L226: "… including the northernmost regions …"
L240-241: CHANGE SENTENCES: "…bodily fluids [15], i.e. S. noctilio larvae are freeze-avoiding."
L249-252: This part reads unclear and confusing. CLARIFY!
L252: "… (Figure 2)."
L264-266: CONDENSE! These sentences sound repetitive!
L271: "Second, the overwintering habit of …"
L272: "… by borers have a …"
L282: "… at temperatures below the SCP …"
L288: "Specifically, in the DM and YS populations …"
Author Response
Response to Reviewer 2 Comments
Dear reviewer:
Thank you for your help, we would carefully modify the article based on your suggestions.
These are specific changes:
TERMINOLOGY:
I suggest the authors use the term "Freezing point" as a synonym of the insects' MELTING POINT throughout the paper and in Fig. 2. This is misleading and incorrect, and should be avoided, especially in the view of non-specialist readers (see [and quote] e.g. Zachariassen K E (1985) Physiology of cold tolerance in insects. Physiol Rev 65, 799-832).
The overwintering insects freeze at the supercooling point (SCP)! The ice crystals formed AFTER freezing (below the SCP) melt at the higher MELTING POINT (MP) during rewarming (thermal hysteresis).
I am missing a description in Methods how these MPs (gray bars in Fig. 2) were determined!
DISCUSSION
The discussion can stand some clarifications here and there.
Thanks for your suggestion, we are so pleasure to learn more knowledge from “Physiology of cold tolerance in insects”. As it said, “The term freezing point is frequently used synonymously with melting point. This is unfortunate, because the freezing point can easily be confused with other important concepts, such as the supercooling point (SCP).” We changed about 13 FREEZING POINT to MELTING POINT and quoted this paper. Such as: “The climate variability hypothesis states that organisms exposed to a wider range of temperature variations exhibit a greater tolerance to extreme temperatures [14]. Insects exposed to temperatures below the humoral melting point (MP) are at risk of fatally freezing their body fluids [15]. “
About Figure.1 please see the attachment.
OTHER:
Include Fig. S1 in the main paper! To me this is a quite interesting and informative recording on the relation of microclimatological thermal conditions between the overwintering hibernaculi and the outside (and historical) temperatures.
Rewrite its legend to (L295-296):
"Figure 1. Mid-winter temperature measured inside and outside the overwintering tunnel, and of the nearest local weather station (Historical)."
Thanks for your suggestion. We used this results as supplementary information, and the information contained in it is to support our discussion. However, we are conducting another experiment with more sample size and more detailed experimental design. We will presnt the results and discuss this topic in detail in a future manuscript.
L28-29: "…that attacks Pinus spp. (Pinaceae). It has become one of the most …
Thanks for your suggestion, we changed it to “Sirex noctilio Fabricius is an invasive woodboring wasp that attacks Pinus spp. (Pinaceae). It has become one of the most important pests in softwood plantations of the Southern Hemisphere [1-5].” (Lines 35-36).
L45: Maybe quoting here (or elsewhere) a recent paper from this journal might be helpful: Käfer et al. (2020) Temperature tolerance and thermal environment of European seed bugs. Insects 11, 197.
Thanks for your suggestion, it helped a lot. We have studied this latest work carefully, and it helped a lot to us. We not only quoted this paper, but some great papers cited by it. Such as L47-48: The climate variability hypothesis states that organisms exposed to a broader range of temperature variations exhibit greater tolerance to extreme temperatures [14]. L264-267: Nonetheless, the temperature is not the only factor limiting larval cold tolerance [14]. According to the climate variability hypothesis and instance, geographical latitude, seasonality in temperature, and also elevation above sea level are thought to correlate with the animals’ temperature tolerance breadth [26-27, see for instance (28-30)].
Reference:
14. Stevens, G.C. The latitudinal gradient in geographical range: How so many species coexist in the tropics. Am. Nat. 1989, 133, 240–256.
26. Sunday, J.M.; Bates, A.E.; Dulvy, N.K. Global analysis of thermal tolerance and latitude in ectotherms. Proc. Biol. Sci. 2011, 278, 1823–1830.
27. Sheldon, K.S.; Tewksbury, J.J. The impact of seasonality in temperature on thermal tolerance and elevational range size. Ecology 2014, 95, 2134–2143.
28. Sunday, J.; Bennett, J.M.; Calosi, P.; Clusella-Trullas, S.; Gravel, S.; Hargreaves, A.L.; Leiva, F.P.; Verberk, W.C.E.P.; Olalla-Tárraga, M.Á.; Morales-Castilla, I. Thermal tolerance patterns across latitude and elevation. Philos. Trans. R. Soc. Lond. B Biol. Sci. 2019, 374, 20190036.
29. Helmut K.; Nikolay S.; Andrea B.; Bettina E.; Arne K. D. S.; Anton S. Temperature tolerance and thermal environment of European seed bugs. Insects, 2020, 11, 197.
30. Oyen, K.J.; Giri, S.; Dillon, M.E. Altitudinal variation in bumble bee (Bombus) critical thermal limits. J. Therm. Biol. 2016, 59, 52–57.
L66-67: The legend text is not correct. Cange it to:
"Figure 1. Sampling sites for S. noctilio in China, showing the geographic range of the species by province. The inset map shows the location in the northeast of China."
Thanks for your suggestion, we changed it to “Figure 1. Sampling sites for S. noctilio in China, showing the geographic range of the species by province. The inset map shows the location in the northeast of China.” at line 86-87.
L68-70: DELETE!
Thanks for your suggestion, we deleted it.
L98-99: CHANGE (here and elswhere) "freezing point (FP)" to "melting point (MP)" (see above)
Thanks for your suggestion, we changed it.
L105-112: CLARIFY: The experimental regime is unclear to me. Eight temperature "gradients"?
Did you cool the larvae to the eight temperatures (or just put them into chambers with those temperatures) and leave them there for the indicated periods of time?
Your understanding is correctly, we cooled the larvae to eight temperatures and placed them there within the specified time.
L118: CHANGE HEADING:
"2.4. Historical temperature data and microclimatological measurements"
Thanks for your help, we changed the heading to "2.4. Historical temperature data and microclimatological measurements"
L121-125: CONDENSE and REWRITE THIS PAPAGRAPH!
"At the sampling sites, the temperature inside overwintering tunnels and the ambient themperature outside were measured in the stem of three Pinus sylvestris which had been injured by S. noctilio. Data (± 1°C) were recorded every ten minutes for the whole winter by electronic thermometers (L91-1, Hangzhou Luge Technology Co. Ltd.; China)."
Thanks for your help, we changed it as your suggestion to " At the sampling sites, the temperature inside overwintering tunnels and the ambient temperature outside were measured in the stem of three Pinus sylvestris, which had been injured by S. noctilio. Data (± 1°C) were recorded every ten minutes for the whole winter by electronic thermometers (L91-1, Hangzhou Luge Technology Co. Ltd.; China). " at line 133-137.
L125-130: TRANSFER THIS PART TO RESULTS!
Thanks for your suggestion, we transferred it to L206-209.
L136: "… date of collection."
Thanks for your suggestion, we changed it to “The SCP data were imported into SPSS 23.0 (IBM, Inc.) and divided into three categories according to the date of collection.”
L137: "… a t test was …"
Thanks for your suggestion, we changed it to “A test of homogeneity of variance was performed for one-way ANOVA to better meet normality criteria, and a t test was performed between each pair of samples to obtain P-values.”
L149‑150: avoid separation of minus sign from number at line break!
Thanks for your suggestion, we rechecked the format to avoid separation of minus sign from number at line break.
L152: insert space after °C
Thanks for your suggestion, we inserted space after °C.
L167-170: CHANGE (here and elswhere) "freezing points (FP)" to "melting points (MP)" (see above)
Thanks for your suggestion, we changed them. L180-183 “The MPs (mean ± SE) of the three overwintering periods were -10.62 ± 0.70°C, -16.06 ± 0.57°C and -11.42 ± 0.59°C, respectively (Figure 2). The mean MP and SCP correlations of all experimental samples were significant (t = 12.4; df =379; P < 0.0001). The MPs corresponded with the SCPs (r = 0.83, P < 0.01) (Figure 2).”
L168-170: UNCLEAR!
correlations with what? The MPs correlated with the SCPs (r = 0.83, P < 0.01) (Figure 2).
Thanks for your suggestion, but according to the last question, we changed it to “The freezing points corresponded with the SCPs (r = 0.83, P < 0.01) (Figure 2).”.
L185: "In contrast, …"
Thanks for your suggestion, we changed it to “In contrast, when larvae were exposed to low temperature for 4 h, less than 30% survived below -20°C.”.
L197; Fig. 3: It would be helpful to have the (color) classes shown in the figure also appearing in the color scale (Mortality: None to All)
Thanks for your suggestion, but we didn’t clear what is “have the (color) classes shown in the figure also appearing in the color scale”.
L195, Table 2: prevent dividing (breaking) of words!
Thanks for your suggestion, we changed the format of Table 2.
Table 2. LLT50 (℃) and LLT99 (℃) of overwintering larvae with different exposure durations.
L199 (and elsewhere): WRITE " (mean ± SE) "
Thanks, we changed it to “(mean ± SE)”, and there is nothing else to be modified after re-checking.
L201: you mean "… Figure 2, …" or "… Figure 4, …"?!
Thanks for your suggestion, we mean "… Figure 4, …", but we didn’t clear what we should modify.
L201-203: THIS IS UNCLEAR, please CLARIFY (cannot see this in Fig. 2 or Fig. 4 clearly)!
Thanks for your suggestion and we are so sorry for that, we mean in Figure 4, and we added it at the end.
L223: "…mean SCP of S. noctilio. However, this does not exclude some extreme values to be lower than the SCP."
Thanks for your suggestion, we changed it to “However, this does not exclude some extreme values to be lower than the SCP.”.
L226: "… including the northernmost regions …"
Thanks for your suggestion, we changed it to “Sirex noctilio is widely distributed, including the northernmost regions of the world and even in the Russian Federation, Canada, and Norway.”.
L240-241: CHANGE SENTENCES: "…bodily fluids [15], i.e. S. noctilio larvae are freeze-avoiding."
Thanks for your suggestion, we changed it to “This finding further indicates that in their natural state, overwintering larvae avoid the freezing of their bodily fluids [15], i.e. S. noctilio larvae are freeze-avoiding.”
L249-252: This part reads unclear and confusing. CLARIFY!
Sorry for the confused we made, we modified the format to describe clearly. “The temperature of JBT only was significantly higher than that in the other regions. During the winter, the environmental temperature and the SCP of geographically distinct populations first decreased and then increased (Figure 2, 4). Nonetheless, the temperature is not the only factor limiting larval cold tolerance.”
L252: "… (Figure 2)."
Sorry for the confused we made, we modified the format to describe clearly. “The temperature of JBT only was significantly higher than that in the other regions. During the winter, the environmental temperature and the SCP of geographically distinct populations first decreased and then increased (Figure 2, 4). Nonetheless, the temperature is not the only factor limiting larval cold tolerance.”
L264-266: CONDENSE! These sentences sound repetitive!
Thanks, we changed it to “Previous analyses of suitable areas are based on the known distribution of a species.”
L271: "Second, the overwintering habit of …"
Thanks, we changed it to “Second, the overwintering habit of S. noctilio should be considered. S. noctilio overwinters in the xylem of plants and undergoes diapause [26], and the tunnels created by borers have a certain insulating effect that reduced the influence of extreme temperatures (Figure S1) [6].”.
L272: "… by borers have a …"
Thanks, we changed it to “Second, the overwintering habit of S. noctilio should be considered. S. noctilio overwinters in the xylem of plants and undergoes diapause [26], and the tunnels created by borers have a certain insulating effect that reduced the influence of extreme temperatures (Figure S1) [6].”.
L282: "… at temperatures below the SCP …"
Thanks, we changed it to “Here, we demonstrate that the overwintering larvae of S. noctilio are freeze intolerant and can only survive for a short period at temperatures below the SCP corresponding to the extremely cold temperatures of several habitats in China.”.
L288: "Specifically, in the DM and YS populations …"
Thanks for your suggestion, but this part had been deleted by the other reviewer’s suggestion.

Round 2
Reviewer 1 Report
The authors addressed the majority of my concerns with the new draft. It is much improved. I have a few minor suggestions that could help improve the manuscript.
1) Lines 39-40: Please remove the sentence “S. noctilio has a strict symbiotic relationship with its symbiotic-fungi Amylostereum areolatum 40 (Fr.) Boidin [5]” It does not fit into the narrative.
2) Line 62: Please rewrite the sentence that starts as “Such as Anoplophora chinensis (Forster) and A. glabripennis (Motschulsky) 62 [17]), Feng et al.”. It is very awkward to read in its present state.
3) Line 65 “Significantly” is spelled wrong.
4) Line 76. Please do not start a sentence with the word “And” it is a conjunction. This sentence could be removed.
5) Data Analysis
I suggest you place the discussion about the Games-Howell test and LSD test before discussing the t-tests so it is clearer they are post hoc tests of the ANOVA. Please rewrite the last sentence of this section. It is not clear what you are trying to say here.
Discussion
6) Line 297-300: It would be helpful if the authors would describe what these additional studies say and how the results of those studies relate to the authors’ findings.
7) I suggest the last paragraph be removed from the paper. It seems like a very interesting project but should be saved for your next publication. the information adds very little to this manuscript.
Author Response
Response to Reviewer 1 Comments
1) Lines 39-40: Please remove the sentence “S. noctilio has a strict symbiotic relationship with its symbiotic-fungi Amylostereum areolatum 40 (Fr.) Boidin [5]” It does not fit into the narrative.
Thanks for your suggestion, we removed it.
2Line 62: Please rewrite the sentence that starts as “Such as Anoplophora chinensis(Forster) and glabripennis (Motschulsky) 62 [17]), Feng et al.”. It is very awkward to read in its present state.
Thanks for your suggestion, we rewrite this sentence as :“Different geographic habitats may cause differences in cold tolerance of insects [such as Anoplophora chinensis (Forster) and A. glabripennis (Motschulsky), see for instance (20-24)]. Feng et al. assessed the variation in cold tolerance of A. glabripennis in geographic populations and between seasons [20,21].”
3) Line 65 “Significantly” is spelled wrong.
Thanks for your suggestion, sorry for our mistake. We rewrite this sentence as :“ The results showed that the SCP and freezing point differed significantly among populations, where the SCP of larvae in the Wulateqianqi (lower environment temperature) population was the lowest and highest in the Beijing(higher environment temperature) population [20].”
4) Line 76. Please do not start a sentence with the word “And” it is a conjunction. This sentence could be removed.
Thanks for your suggestion, we removed this sentence.
5) Data Analysis
I suggest you place the discussion about the Games-Howell test and LSD test before discussing the t-tests so it is clearer they are post hoc tests of the ANOVA. Please rewrite the last sentence of this section. It is not clear what you are trying to say here.
Thanks for your suggestion, we advanced the discussion about the Games-Howell test and LSD test from L173-176 to L169-172 and rewrite the last sentence as “Spearman’s method was used to analyse the correlation between the characteristic values of ambient temperature and SCP (because the characteristic values of ambient temperature were continuous and had a large order of magnitude difference compared with SCP).”
Discussion
6)Line 297-300: It would be helpful if the authors would describe what these additional studies say and how the results of those studies relate to the authors’ findings.
Thanks for your suggestion, we rewrite this part at L262-271, as “Based on our results, the temperature of JBT was significantly higher than that in the other regions, then YS [Figure 4]. Although the average temperature of YS during the whole wintering period was similar to that of HG and DM, its temperature suddenly droped in the middle of the wintering period, which was also consistent with the situation shown by SCP. The SCPs of each population corresponded to the sampling site temperature during the overwintering period (JBT > YS > DM > HG), the northern populations (HG and DM) were considerably more cold hardy than were the southern populations (JBT and YS), as indicated by the SCP measurements. The average SCP of all population during the overwintering period was consistent with the average site temperature during the overwintering period (Pre-winter and Post-winter > Mid-winter).”
7) I suggest the last paragraph be removed from the paper. It seems like a very interesting project but should be saved for your next publication. the information adds very little to this manuscript.
Thanks for your suggestion, we removed it.
